# AvianLexiconAtlas: A database of descriptive categories of English-language bird names around the world

Erin S. Morrison[1]*, Guinevere P. Pandolfi[1], Stepfanie M. Aguillon[2], Jarome R. Ali[3], Olivia Archard[1], Daniel T. Baldassarre[4], Illeana Baquero[1], Kevin F. P. Bennett[5], Kevin M. Bonney[1], Riley Bryant[1], Rosanne M. Catanach[6], Therese A. Catanach[7], Ida Chavoshan[1], Sarah N. Davis[8], Brooke D. Goodman[4], Eric R. Gulson-Castillo[9], Matthew Hack[9], Jocelyn Hudon[10], Gavin M. Leighton[11], Kira M. Long[12]¤, Ziqi Ma[1], Dakota E. McCoy[13], J. F. McLaughlin[14], Gaia Rueda Moreno[1], Talia M. Mota[1], Lara Noguchi[1], Ugo Nwigwe[1], Teresa Pegan[9], Kaiya L. Provost[15], Shauna A. Rasband[5], Jessie Frances Salter[16], Lauren C. Silvernail[17], Jared A. Simard[1], Heather R. Skeen[18], Juliana Soto-Patiño[19], Young Ha Suh[20], Qingyue Wang[1], Matthew E. Warshauer[21], Sissy Yan[1], Betsy Zalinski[1], Ziqi Zhao[1], Allison J. Shultz[20]

1 Liberal Studies, New York University, New York, New York, United States of America, 2 University of California, Los Angeles, California, United States of America, 3 Princeton University, Princeton, New Jersey, United States of America, 4 SUNY Oswego, Oswego, New York, United States of America, 5 University of Maryland, College Park, College Park, Maryland, United States of America, 6 Trinity, Alabama, United States of America, 7 Academy of Natural Sciences of Drexel University, Philadelphia, Pennsylvania, United States of America, 8 Carnegie Museum of Natural History, Pittsburgh, Pennsylvania, United States of America, 9 University of Michigan, Ann Arbor, Michigan, United States of America, 10 Royal Alberta Museum, Edmonton, Alberta, Canada, 11 SUNY Buffalo State University, Buffalo, New York, United States of America, 12 University of Idaho, Moscow, Idaho, United States of America, 13 University of Chicago, Chicago, Illinois, United States of America, 14 University of Alaska Anchorage, Anchorage, Alaska, United States of America, 15 Adelphi University, Garden City, New York, United States of America, 16 Santa Monica College, Santa Monica, California, United States of America, 17 Oswego, New York, United States of America, 18 Negaunee Integrative Research Center, Field Museum of Natural History, Chicago, Illinois, United States of America, 19 University of Illinois at Urbana-Champaign, Urbana, Illinois, United States of America, 20 Ornithology Department, Natural History Museum of Los Angeles County, Los Angeles, California, United States of America, 21 New York, New York, United States of America

These authors contributed equally to this work.
¤Current address: Center for Conservation Genomics, Smithsonian's National Zoo and Conservation Biology Institute, Washington, DC, United States of America
* erin.morrison@nyu.edu

## Abstract

Common names of species are important for communicating with the general public. In principle, these names should provide an accessible way to engage with and identify species. The common names of species have historically been labile without standard guidelines, even within a language. Currently, there is no systematic assessment of how often common names communicate identifiable and biologically relevant characteristics about species. This is a salient issue in ornithology, where common names are used more often than scientific names for species of birds in written and spoken English, even by professional researchers. To gain a better understanding of

**Data availability statement:** The data, glossary, and gazetteer reported in this article can be accessed at https://github.com/ajshultz/AvianLexiconAtlas.

**Funding:** Funding was provided by the New York University Liberal Studies New Faculty Scholarship Award to E.S.M. The funders had no role in study design, data collection and analysis, decision to publish, or preparation of the manuscript.

**Competing interests:** The authors have declared that no competing interests exist.

the types of terminology used in the English-language common names of bird species, a group of 85 professional ornithologists and non-professional contributors classified unique descriptors in the common names of all recognized species of birds. In the AvianLexiconAtlas database produced by this work, each species' common name is assigned to one of ten categories associated with aspects of avian biology, ecology, or human culture. Across 10,906 species of birds, 89% have names describing the biology of the species, while the remaining 11% of species have names derived from human cultural references, human names, or local non-English languages. Species with common names based on features of avian biology are more likely to be related to each other or be from the same geographic region. The crowdsourced data collection also revealed that many common names contain specialized or historic terminology unknown to many of the data collectors, and we include these terms in a glossary and gazetteer alongside the dataset. The AvianLexiconAtlas can be used as a quantitative resource to assess the state of terminology in English-language common names of birds. Future research using the database can shed light on historical approaches to nomenclature and how people engage with species through their names.

## Introduction

Humans have observed and classified living organisms for thousands of years, across both cultures and languages [1]. The names of species can have consequences for how they are perceived by people, and this can, in turn, affect how people engage with species as part of education, research, and conservation efforts [2–6]. Constructing classification systems that delineate and name organisms can be subjective [5,7,8]. To standardize scientific animal names, the International Commission on Zoological Nomenclature (ICZN) has overseen the assignment of a unique universal scientific name for each recognized animal species since 1895. This scientific name is written as a binomial, often in Latin, and includes the genus and species names in the Linnaean taxonomy system [9]. The vernacular names of species, however, are not standardized across languages and have been labile across history, even within a single language or region [10–12]. Known as 'common' or, as argued by [13], 'standard' names, the use of these vernacular names can communicate information about species in a way that is more accessible to a wider audience outside of the scientific community [3,14,15]. However, even these common names may be unique to a single region, species that span cultures may take on multiple common names in the same language, or the same common name may be unofficially used for multiple species.

In ornithology, the common names of birds tend to be used more frequently in spoken and written English than their scientific binomial names, even among professionals [16]. Despite this, the English-language common names of birds are only standardized regionally by professional ornithology societies [17], and there is no universal set of rules for English-language common names [3,18–21]. Within the

ornithological community, there is much reflection about what the terminology in standardized English-language common names should communicate about species [5,16,22–24]. Currently, some names directly describe characteristics of a species (e.g., Yellow-rumped Warbler, *Setophaga coronata*), while other names are ambiguous (e.g., Barnacle Goose, *Branta leucopsis*) or unrelated to the species' biology (e.g., Wilson's Warbler, *Cardellina pusilla*). To understand what English-language common names currently communicate about species of birds, it is necessary to comprehensively examine the scope and variability of the terminology used in these names.

While the history of English-language bird names has been extensively documented [25–28], there is currently no systematic resource of the types of terminology used in the English-language common names of birds. We therefore endeavored to inventory and classify the terms currently used as the specific descriptors in English-language common names of all avian species and assembled these data into a freely available database known as the AvianLexiconAtlas. In the database, the unique descriptor of each species name is assigned to one of ten distinct categories associated with aspects of avian physical traits, avian natural history, or human culture. To validate the utility of the dataset as a resource for professional ornithologists, linguists, and the general public to learn more about how humans name species, we summarized frequency differences among different categories across species, and also examined trends in the terminology associated with species' close taxonomic relationships or shared biogeography. We use the results from these descriptive analyses to propose directions for future quantitative research using the database that could focus on the relationship these human-constructed names have with aspects of avian biology or examine historical trends in the terminology.

Additionally, since one goal of standardized common names is to be useful for the general public, data collection for this project presented a unique opportunity to involve data collectors from both within and beyond the academic ornithology community. The inclusion of undergraduate students and amateurs as data collectors was a chance to assess familiarity with the terminology used in English-language bird names. During data collection, data collectors were also asked to contribute to a glossary and a gazetteer when they encountered unfamiliar terminology. We make these resources and the dataset produced from their work available for future analyses on the state of avian nomenclature as the AvianLexiconAtlas.

## Materials and methods

### Ethics statement

The New York University Human Research Protection Program (HRPP) determined that the work involved in the data collection for the database did not meet the criteria involving human subjects per the United States' regulations for the protection of human subjects (45CFR46.102(e)) [29]. This verbal and written determination was made because the focus of the research was not on collecting information about the data collectors, but only involved a crowdsourcing model where the data collectors helped categorize the meanings behind unique descriptors in bird names. As such, the data collection methods did not require HRPP review and approval.

### Selection of the eBird/Clements checklist for species names

The English-language common names in the database are based on the taxonomy of 10,906 species in the 2022 eBird/Clements Checklist [30]. At the time of the data collection, there were two additional annually curated global checklists with similar, but not identical, taxonomy to the eBird/Clements Checklist. The International Ornithological Congress (IOC) Checklist 12.2 contained 11,140 species, and 86.9% of these species had the same scientific and common names in the eBird/Clements 2022 Checklist [20]. The Handbook of Birds of the World (HBW) and BirdLife International Checklist 7.0 contained 11,170 species, and 79.7% of these species had the same scientific and common names in the eBird/Clements 2022 Checklist [31]. We made the decision to use the 2022 eBird/Clements checklist based on its alignment with the comprehensive *Birds of the World* online database [32], the main reference source for the data collection. The eBird/Clements

taxonomy used in *Birds of the World* incorporates the HBW and BirdLife International taxonomy but makes more conservative distinctions between species and subspecies (refer to [33] and [34] for more details).

## Establishment of categories for species' English-language common names

A majority of bird species' English-language common names contain a unique descriptor followed by a name it shares with closely related species. For example, the unique descriptors in Common Ostrich (*Struthio camelus*) and Somali Ostrich (*Struthio molybdophanes*) are 'Common' and 'Somali', respectively, while the shared name is 'Ostrich'. Only the unique descriptor in a species' English-language common name, not the shared name, was categorized for the database.

Prior to the start of data collection, we ran a series of trial categorizations with a random sample of 30 species drawn from the eBird/Clements 2021 Checklist [35]. We started with a list of potential categories and three people independently attempted to categorize each of the species names without communicating with each other. Importantly, this group included two professional ornithologists and one undergraduate student without any formal training in ornithology. As a result of this trial, we established via consensus 10 categories for the unique descriptors that align with independent categorization for both common and scientific bird names [26,27] and encompass aspects of avian physical traits (physical trait in both sexes, male-only physical trait, female-only physical trait, size), avian natural history (behavior, geographic location, natural history), and human-centered terminology independent of the biology of the species in the English language (local language, miscellaneous, eponyms (named after a particular person)) (Table 1).

## Categorization of species common names

**Data collectors.** A total of 85 people participated in the data collection for the AvianLexiconAtlas. Of these participants, 56 were pursuing undergraduate degrees (students), 7 had at least an undergraduate degree but no formal training in ornithology (amateurs), and 22 were pursuing graduate degrees or worked in post-graduate careers related to ornithology (professionals). Of the 56 undergraduate students, 50 were enrolled in an introductory biology seminar course for non-science majors at New York University during the Fall 2022 semester. All data collectors who categorized at least

**Table 1. Categories used to group unique descriptors in English common names of bird species.**

| Category | Description |
|---|---|
| **Avian physical traits** | |
| Both sexes physical trait | Physical characteristics that are observable in both males and females of a species. Includes: size; colors and patterns of feathers, skin, beak, feet, and eyes; general terms related to plumage complexity (e.g.,: painted, beautiful, handsome, drab, ornate, plain). |
| Male physical trait | Physical characteristics that are only observable and clearly identifiable in males of a species. |
| Female physical trait | Physical characteristics that are only observable and clearly identifiable in females of a species. |
| Size | Relative physical size of the species (e.g.,: fairy, giant, greater, lesser, little, pygmy). |
| **Avian natural history** | |
| Behavior | Specific behavior associated with the species (e.g.,: vocalizations, personality characteristics such as shy or cryptic, or named after another species that has a similar behavior). |
| Geographic location | Specific geographic location. Includes: general geographic descriptors (eastern, western); regions named after local cultures or communities. |
| Natural history | Relates to habitat (e.g.,: mountain, grassland, highland, upland, lowland, paradise), nest type, diet of the species, smell, or nest location. |
| **Human-centered terminology** | |
| Local language | Phonetic English pronunciation of species name in a non-English language. |
| Miscellaneous | Descriptor that does not clearly fit into any of the other categories. |
| Person | Name of a person. |

100 species were eligible for authorship credit provided that they agreed to review the manuscript prior to submission for publication. Students in the course were given in-class participation credit for engaging in the data collection during two 75-minute class sessions. This participation grade counted towards 1.6% of their final grade.

**Data collection.** Data collectors were split into two groups (A and B), and each group evaluated identical lists of 10,906 species. Each species name was therefore independently categorized by two different people. Each data collector only worked on one of the two datasets and was not given access to the other dataset. Given the large number of people involved in the data collection, categorizing the names of species in duplicate was built into the methodology in order to be able to validate the accuracy of each species' category assignment and identify descriptors that were difficult to classify. Professionals, amateurs, and the six students not enrolled in the introductory biology course were randomly assigned to Dataset A or B based on the order in which they signed up for the project. The introductory biology course consisted of two sections of 25 students each, and each of these sections was assigned to a different dataset. A total of 39 individuals worked on Dataset A (28 students, 2 amateurs, and 9 professionals) and a total of 47 individuals worked on Dataset B (29 students, 5 amateurs, and 13 professionals). Data collectors signed up for 10 species at a time, partitioned by the order they were listed in the 2022 eBird/Clements Checklist. Groups of 10 species had to be categorized in sequential order and data collectors were not allowed to selectively choose groups to categorize.

Categorization of the unique descriptor in a species' name first started by finding its account in the *Birds of the World* database and then the category assignment associated with the unique descriptor was determined using the provided text, images, and media in the species account. If the origin of the unique descriptor could not be determined from *Birds of the World*, data collectors would then conduct a broader internet search to find other potential sources. Any sources used outside of *Birds of the World* were recorded in the complete dataset. If the origin of the unique descriptor could not be determined based on information in *Birds of the World* or a subsequent internet search within 10 minutes, data collectors were asked to note that they could not identify the name within the given time limit and move on to the next species in their list.

To facilitate categorizations for the data collectors who were not familiar with common terms in ornithology, a glossary was developed for the project that included diagrams of common field markers and definitions of common ornithology descriptors of traits and colors. Participants were granted editing access to the glossary (S1 Appendix), and were asked to add additional terms and definitions they had to look up during the data collection period. A gazetteer (S1 Appendix) that included descriptions of all geographic locations encountered in species' names was also added to this glossary as a reference. In addition to these resources, data collectors were provided with two lists of eponyms documented in species' English-language names compiled by Beolens and Watkins [25] and Bird Names For Birds [24].

**Dataset reconciliation.** When the two duplicate datasets were compared, 906 out of the 10,906 species (8.4%) were assigned to different categories in the two datasets (mismatches). There were also 9 additional species that were designated as time limited by data collectors in both datasets. The majority of the 906 species that were assigned to different categories in Datasets A and B were labeled as a physical trait appearing in both sexes in one dataset and as a male-only physical trait in the other dataset (S1 Fig). This was followed by mismatches in category assignments between geographic location and life history, general descriptor and a physical trait in both sexes, and life history and behavior. These 915 species were subsequently re-categorized independently in three identical datasets (C, D, E) by 10 professional ornithologists that had also participated in the first round of data collection, each of whom was assigned to one of the three datasets. When the 915 species' re-categorizations were compared across the three datasets, categories for 554 species (60.5%) matched across all 3 datasets and were retained for the final dataset. Categories for 324 (35.4%) of the re-categorized species matched in 2 out of 3 datasets, and the majority category was retained for these species in the final dataset. There was no category agreement across the three datasets for the remaining 37 (4.0%) re-categorized species. The final category for each of these species was jointly assigned by the lead authors of the study (ESM and

AJS), together, based on the references provided by all those who had categorized the species in the first and second rounds.

### Data validation

To validate the utility of the AvianLexiconAtlas dataset, we summarized the frequency of unique descriptors in species' names and also examined phylogenetic and biogeographic categorization trends. All analyses were conducted in R 4.3 [36], and only considered the first word in the descriptor of species' English-language common names. The R package wordcloud2 0.2.1 [37] was used to generate visualizations of the relative frequencies of terminology used in the name descriptors of species across the dataset and across each of the three general categories.

To examine categorization trends among species across taxonomic groups in the database, we mapped the general categories assigned to each species' name (i.e., avian physical traits, avian natural history traits, or human-centered terminology; Table 1) to a time-calibrated maximum clade credibility global avian phylogeny of 10,824 species [38]. The taxonomy in this phylogeny is based on the eBird/Clements 2021 checklist. The R package *clootl* 0.0.0.900 [39] was used to extract the phylogeny, and the final tree was pruned to only include the 10,775 species with common names that occur in both the 2021 and 2022 checklists (S3 Fig.). We measured phylogenetic signals for each of these categories using Fritz & Purvis' $D$ for binary traits (presence/absence) [40], using the package *caper* 1.0.3 [41] in R. $D$ calculates the number of sister-clade differences in a binary trait for a given phylogeny [38] (S1 Table). An estimated $D$ close to 1 represents a random distribution of a binary trait among related species on the phylogeny, while an estimated $D$ close to 0 represents a clumped distribution of a binary trait among related species that would be expected under the Brownian motion model of evolution [40]. To test for significance, $D$ was estimated for two different trait distributions for each trait that were simulated on the tips of the same phylogeny based on (1) randomly reshuffling the trait values and (2) trait evolution under Brownian motion. Each simulation was repeated 1,000 times.

To identify any geographic categorization trends, we mapped all species names to the IOC World Bird List 13.2 [42]. We extracted the "general region" for the breeding range of each species from the range description data included in the IOC World Bird List 13.2 (S1 Appendix). The IOC World Bird List compiles range descriptions for each species based on several authoritative sources and classifies these geographic ranges into general regions, which are generally at the subcontinent level, but also include a separate classification for primarily oceanic species [43]. Species with a breeding range in more than one general region were assigned to each region. We acknowledge the importance of the non-breeding range, but the lack of standardization of non-breeding range information across species precluded our ability to assign species to a general region for non-breeding range. We grouped all oceans together into a single category to increase interpretability, dropped species with a Worldwide distribution (n = 10), an Antarctic distribution (n = 12; sample size too small for analyses), and 110 species that could not be mapped due to taxonomic differences, for a total of 12,058 species-general region assignments in this dataset. To test whether general regions varied in the proportions of general descriptor categories (avian physical traits, avian natural history traits, or human-centered terminology), we conducted a Chi-square test using the chisq.test function in R.

## Results

### Database implementation

The AvianLexiconAtlas database consists of a dataset with the final consensus categorization of the unique descriptor in the English-language common name for all 10,906 species of birds in the eBird/Clements 2022 taxonomy checklist as well as the comprehensive glossary and gazetteer compiled by the contributors to the data collection (S1 Appendix). The database also includes the raw dataset that documents all category assignments for each species' name across the duplicate datasets (A and B) from round 1 for all 10,906 species and the triplicate datasets (C, D, E) from round 2 for the 915 species without a consensus category in round 1. Downloadable CSV files of the datasets (final decisions and raw data)

and PDF versions of the glossary and gazetteer are freely accessible on the AvianLexiconAtlas GitHub site (S1 Appendix, https://github.com/ajshultz/AvianLexiconAtlas). To allow for different modes of access to the database information, the GitHub site also contains read-only links to Google Sheets and Google Docs versions of both of the datasets and the Glossary and Gazetteer document, respectively. With the establishment of the protocol for name categorization, our long-term vision for the database is that it will continue to track annual revisions to taxonomy and English-language common names in the eBird/Clements taxonomy through a yearly database update.

### Dataset description

Of the 10,906 species categorized, 57% are named based on avian physical traits, 32% are named based on avian natural history, and 11% are named based on human-centered terminology unrelated to the biology of the species (Fig 1; Table 2; S1 Appendix). Within avian physical traits, most species are named after traits present in both sexes, although nearly 1,000 species are named after traits only found in males, whereas only 20 species are named after traits only found in females (Fig 1; Table 2). Within avian natural history, most species are named after their geographic location compared to other natural history attributes or behavior (Fig 1; Table 2), and in human-centered terminology, most species are named after people (eponyms), compared to local language or miscellaneous categorizations (Fig 1; Table 2). There is variation across categories for the number of times a term was repeated across species, with local language having the greatest proportion of distinct terms, and size having the smallest proportion of distinct terms (Table 2).

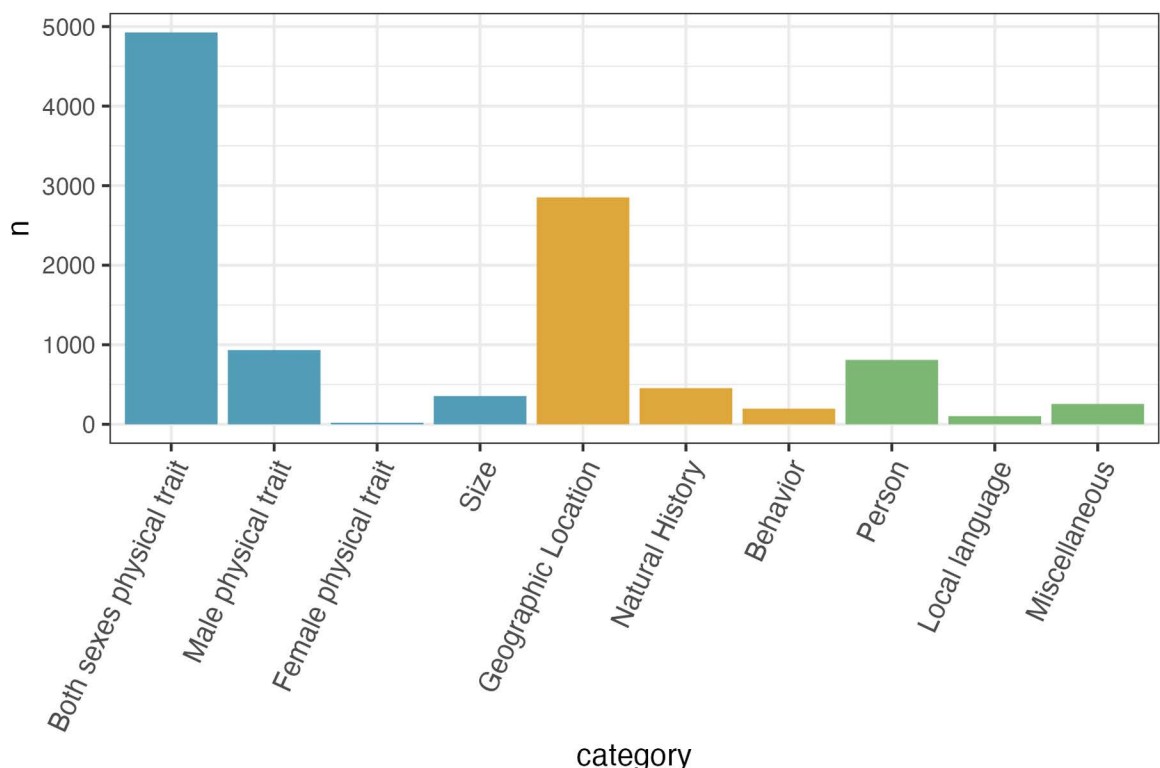

**Fig 1. Counts of descriptor categories of bird common names in English-language per category in the final dataset of 10,906 bird species.** Categories are colored by avian physical traits (blue), avian natural history (orange), and human-centered terminology unrelated to the biology of a species (green).

**Table 2. Numbers and proportions of unique descriptors as common names in different categories. Calculations for the mean and median frequencies of unique descriptors only include descriptors that occur more than once among species names in each category.**

| Category | General category | N species | N unique descriptors | Proportion unique descriptors | Mean frequency of unique descriptors | Median frequency of unique descriptors |
|---|---|---|---|---|---|---|
| Both sexes physical trait | Avian physical trait | 4926 | 1530 | 0.31 | 6.30 | 3 |
| Male physical trait | Avian physical trait | 933 | 517 | 0.55 | 3.62 | 3 |
| Female physical trait | Avian physical trait | 20 | 19 | 0.95 | 2.00 | 2 |
| Size | Avian physical trait | 355 | 28 | 0.08 | 22.80 | 13 |
| Geographic location | Avian natural history | 2852 | 739 | 0.26 | 7.02 | 4 |
| Natural history | Avian natural history | 455 | 196 | 0.43 | 4.32 | 2 |
| Behavior | Avian natural history | 197 | 145 | 0.74 | 3.00 | 2.5 |
| Person | Human-centered terminology | 810 | 505 | 0.62 | 3.19 | 2 |
| Local language | Human-centered terminology | 102 | 101 | 0.99 | 2.00 | 2 |
| Miscellaneous | Human-centered terminology | 256 | 112 | 0.44 | 7.26 | 4 |
| Total dataset | | 10906 | 3499 | 0.32 | 5.89 | 3 |

We next investigated the frequencies of the name descriptors (for this purpose, we considered the first word in the name as the descriptor) across the dataset and across categories. The five most common descriptors in the dataset are "African" (69 times), "Common" (68 times), "Lesser" (65 times), "Great" (65 times), and "Black" (64 times) (Fig 2). We note that after the size-related terms, many of the commonly used physical trait descriptors are related to plumage color (e.g., black, white-browed, spotted) (S2 Fig). The natural history-related common terms are geography related, especially regarding cardinal directions (e.g., northern, southern) (S2 Fig). The human-centered terminology has very few commonly used terms, except for the term "Common" (S2 Fig).

## Phylogenetic and geographic categorization trends in the dataset

All three general categories of species names exhibit weak phylogenetic signal across 10,775 species (Figs 3, S3, S1 Table). Species names associated with avian physical traits ($D = 0.729$) and avian natural history traits ($D = 0.745$) have somewhat stronger phylogenetic signal than names associated with human-centered terminology ($D = 0.875$). Based on simulation tests of the trait distributions, the phylogenetic signal $D$ of each category differs significantly ($P < 0.001$) from what would be expected under Brownian motion ($D = 0$), but also differs significantly ($P < 0.001$) from what would be expected from a random trait distribution on the phylogeny ($D = 1$) for each category.

The proportion of general descriptor categories is significantly different across geographic regions (Fig 4; $\chi^2 = 940.43$, df $= 14$, $P < 0.0001$). South America and Middle America show the highest proportions of species named after avian physical traits (0.74 and 0.72, respectively), and Oceans show the lowest proportion (0.26). Oceans show the highest proportion of species named after avian natural history (0.58) and Middle America and South America show the lowest (0.20 and 0.19, respectively). Africa and Oceans show the highest proportions of species named after human-centered terminology (both 0.17) and South America shows the lowest (0.06).

## Discussion

The AvianLexiconAtlas contains assignments for the descriptive English-language common names of 10,906 bird species into 10 categories. In addition to the category assignments themselves, the AvianLexiconAtlas contains all references

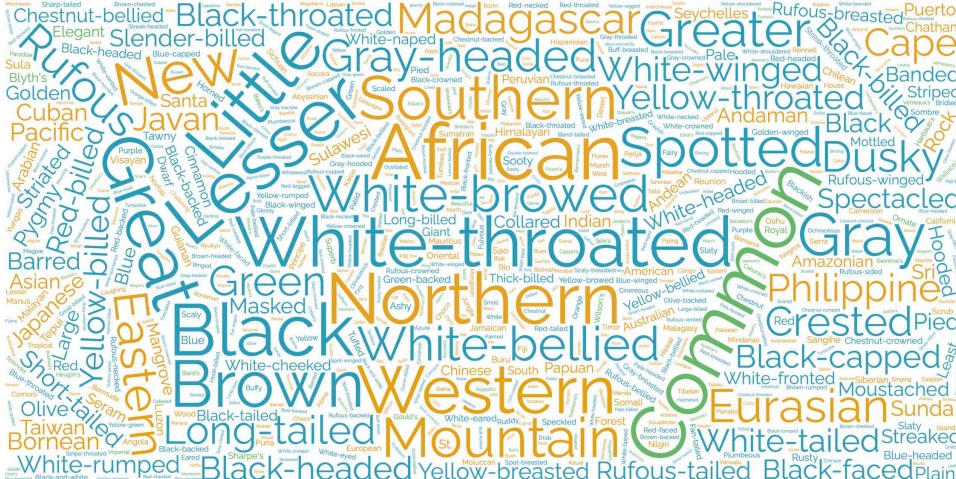

**Fig 2. Word cloud of most common descriptors across the dataset.** Word sizes are scaled to the number of times they are repeated, and descriptors are colored according to their general category: avian physical trait (blue), avian natural history trait (orange), and human-centered terminology (green).

for the assignments based on information found outside of the *Birds of the World* database, a glossary of common terminology and a gazetteer of geographic locations used in English-language common names. We find that 89% of species are named after some aspect of their biology, whether it be their appearance, ecology, behavior, or geographic location. We establish that the most common way to name species is after a physical trait, usually a physical trait that is found in both sexes. We also find a tendency for the names of species that are related to each other or from similar geographic areas to share similar types of descriptors. The assembly of this database, which was a successful collaboration among professional ornithologists, amateurs, and students, has also led to insights about the nature of term recognition and the difficulty, in some cases, of understanding the meaning behind descriptors, including some that many professional ornithologists would have considered to be general knowledge. These initial observations establish the utility of the AvianLexiconAtlas for future research into historical and biological patterns in avian nomenclature, which we will outline here.

We find that physical traits are the most frequently applied category, and one potential explanation could be centuries of specimen-based taxonomic descriptions. Many species were originally formally described primarily by scientists working with specimens in collections, whether in museums or private collections. Specimens themselves typically only have information about physical traits and size, though well-curated metadata should also reveal geographic location, natural history, and, on occasion, behaviors. If the original historical basis of species taxonomy is specimen-based, future work using the database might investigate whether both scientific nomenclature and English-language nomenclature are biased toward descriptions from specimen-associated information.

An additional observation from the database is that the frequency with which types of physical trait descriptors are used in common names in the database corresponds to documented patterns of avian trait diversity. Among species that are named after their physical traits, the majority have common names that describe a trait that occurs in both sexes (Fig 1). This outcome aligns with documented evidence that most bird species do not display any sex differences in plumage brightness and/or pattern [44]. Most common names that do describe sex-specific traits of species, however, refer to traits that only occur in the males, and rarely identify traits unique to the females (Fig 1). One potential reason for this discrepancy is the tendency for humans to focus on more elaborate features that are more commonly found in males, which has been proposed as one explanation for the biases towards male specimens in museum collections [45]. Another explanation for the bias towards naming species after traits found only in males is that, due to intersexual selection, many males

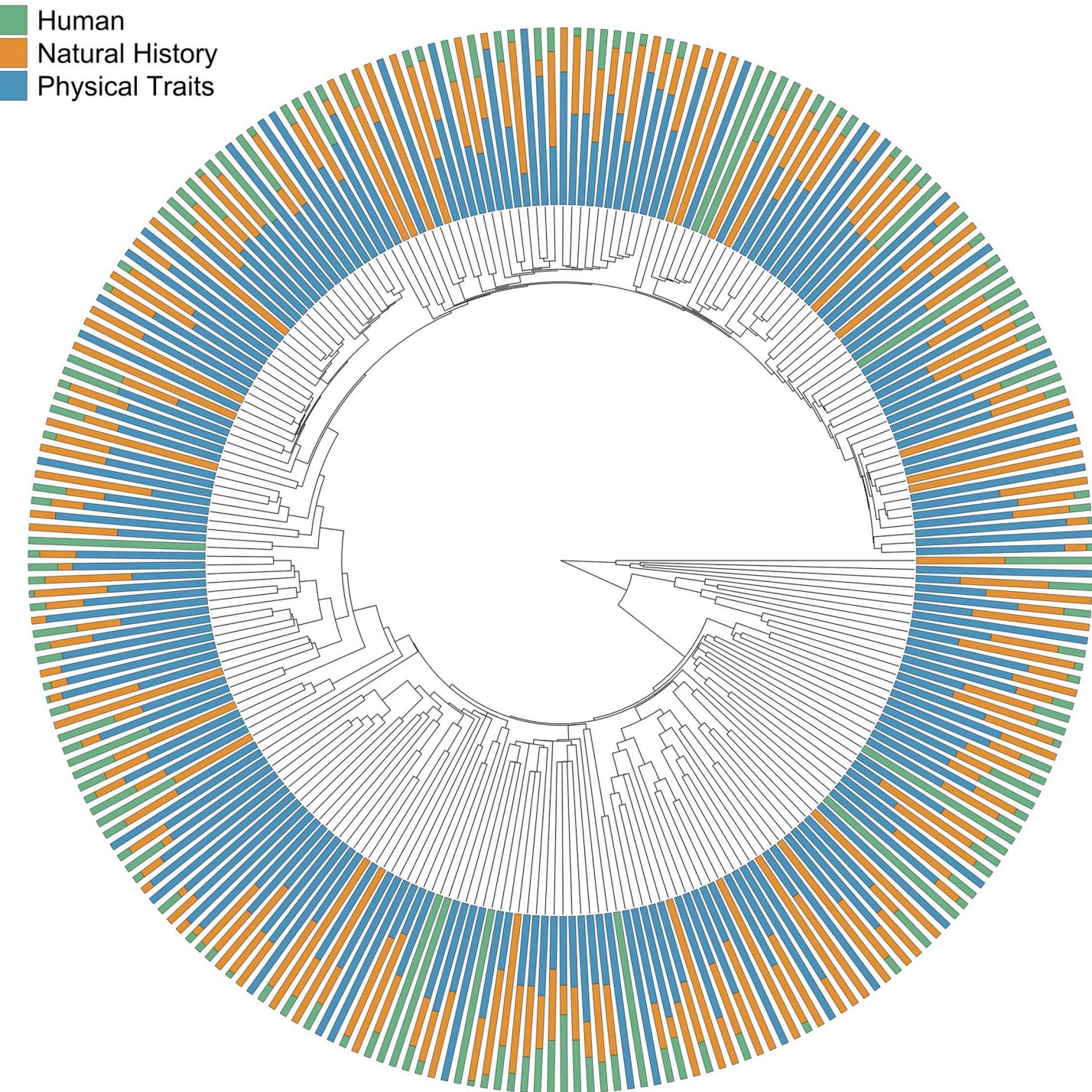

**Fig 3. Cladogram of 249 families of birds.** The location of each family branch was determined by a representative species in a time calibrated phylogeny of 10,775 species of birds [38]. At the tips, the shaded lines represent the proportion of species in each family with English common names associated with the general categories avian physical traits (blue), avian natural history (orange), and human-centered terminology unrelated to the biology of the species (green) (Table 1). See S3 Fig for species-specific name categories mapped to phylogenetic relationships.

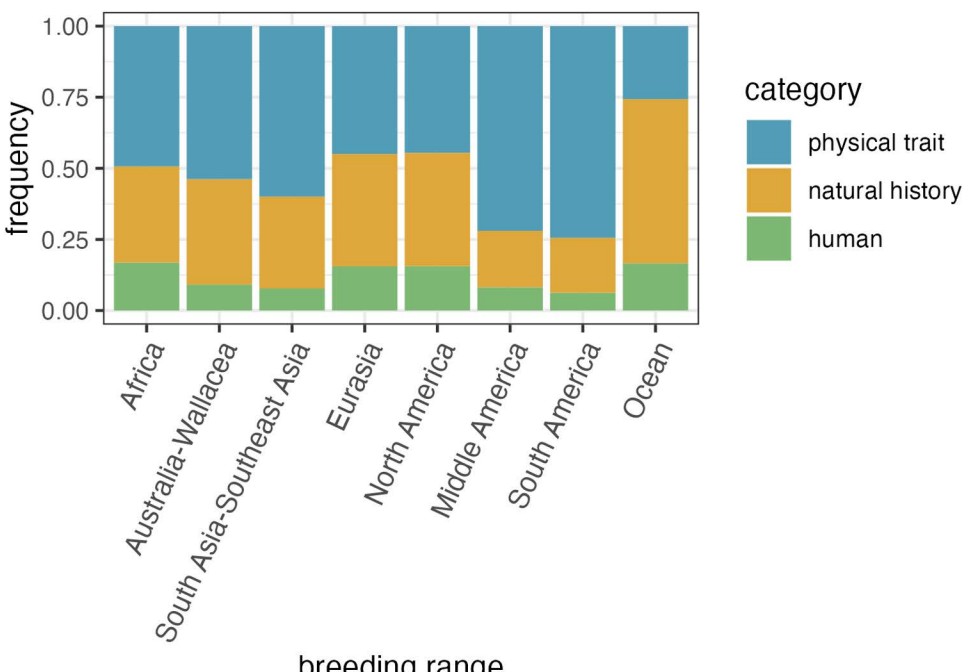

**Fig 4. The frequency of general descriptor categories in species' breeding ranges.** General categories are avian physical traits (blue), avian natural history (orange), and human-centered terminology unrelated to the biology of a species (green).

have more elaborate and faster evolving physical traits than females in dichromatic species [44,46–49]. Males of species in lineages characterized by a large degree of sexual dichromatism may have more unique and prominent traits that can be used as descriptive identifiers compared to females of the same lineage. Female plumage color does vary across species within lineages, but it tends to converge on cryptic colors and patterns that are difficult to distinguish, and thus these traits may not be as easy to use for identification markers between species [46,50–52]. Future work using the database could compare naming patterns to existing data on the degree of sexual dimorphism in species [44,46,53]. It is likely that species that are named after single-sex traits will be more sexually dimorphic than species that are named after both-sex traits, but it remains to be seen whether this pattern would hold for species named after behaviors or other potentially less sex-specific aspects of biology.

When we investigated whether the names of related species tend to be associated with the same categories, we found that avian-centered name categories have relatively stronger phylogenetic signals than human-centered name categories. This result aligns with previous work establishing that the types of traits used to name species often have phylogenetic signals, including color [54–57], song [58–60], and ecology [48,61,62]. Anecdotally, participants noticed that closely-related species have similar naming schemes, particularly in species-rich groups found in the Neotropics. For example, in the 45 species of *Grallaria* antpittas, 22 and 18 species are named after physical traits or geographic locations, respectively. This is also demonstrated by geographic trends, which show that the large majority of species in Middle and South American geographic regions are named after physical traits. Conversely, oceanic species are most commonly named after their natural history (primarily geographic location), likely reflecting that many ocean-dwelling species are difficult to tell apart by plumage or other morphological traits, but live in different geographic areas. Future work using the database could further investigate fine-scale variation in specific clades (i.e., orders or families) or geographies.

In addition to the category assignments, the use of crowdsourcing and duplicate datasets in the data collection for the database also provided an important perspective into the accessibility of English-language common names. We found that most of the mismatches between the duplicate datasets arose because, in some cases, it is difficult to distinguish whether a physical trait occurs in both sexes of a species or only in males (S2 Fig). In the Maroon Pigeon (*Columba thomensis*), for example, parts of the male's plumage are described in *Birds of the World* as "rich maroon", while female coloration is described as "duller… with only hints of maroon", and this distinction is difficult to make out in the provided images [63]. For issues such as this, we had to decide the extent that a trait would have to occur in both sexes for it to be assigned to that category, and much of this was up to personal interpretation. The resolution to independently categorize the mismatched species again in triplicate was an opportunity for consensus agreement in most species, but disagreements across all three datasets still remained for some species' names. Another issue that likely contributed to category mismatches was the use of historical and specialized English-language terminology in species names. Diagrams of the names and locations of parts of the bird were provided to all data collectors at the start of the project, but it was the terminology used to describe colors and patterns that proved to be one of the biggest issues (for example, "painted", "festive", or "glowing"). These types of descriptors comprise a majority of the words in the glossary that all participants were asked to contribute to when they encountered an unknown term. Similarly, the gazetteer was started during the first round of data collection when both professional and non-professional participants struggled to find information on outdated or rarely used names for geographic locations. This brings up a question of how useful specialized terms for color, pattern, and geographic location are in English-language common names, even if they are very specific descriptors, if many people do not know what they mean without further research. The English language is constantly evolving at regional and global scales [64,65] and the way colors are categorized and described can vary among people [66]. Furthermore, the diversity of color terms observed in the dataset are likely associated with how often general categories of color (white/black, red, green/yellow, blue, brown, purple/pink/orange/grey) are referenced across species [67,68]. These issues highlight the lack of consistency that is inherent in many of the trait descriptors used in common names, and could be analyzed as a type of data in and of itself in the future. For example, further examination into the types of terminology used in the 915 species' names that required a second round of categorization as well as the terms listed in the glossary and gazetteer may be useful for current discussions surrounding changes to some English-language common names.

The scope of this database represents the first comprehensive source for further quantitative examinations of the types of terminology used in the common names of birds. This provides a starting point for future development of the AvianLexiconAtlas. First, the categories were intentionally designed to broadly capture distinct themes in the terminology used for English-language common names, with a particular focus on comparing how often terminology is associated with different properties of avian biology or human culture. Future expansions of the AvianLexiconAtlas database could expand these contents into more specific subcategories. For example, Burrowing Owls (*Athene cunicularia*) and Barking Owls (*Ninox connivens*) are both named after behaviors, but the former describes nesting habits and the latter describes vocalizations. Likewise, Golden-crowned Sparrows (*Zonotrichia atricapilla*) and White-throated Sparrows (*Zonotrichia albicollis*) are both named after physical traits (both sexes), specifically color, but the colors themselves and body parts are different. Striped Sparrows (*Oriturus superciliosus*), on the other hand, are named after a pattern that does not appear to be specific to any given body part. Now that the AvianLexiconAtlas has established which English-language common names are associated with avian biology, future research can provide insight into what specific types of biological descriptions are used in these common names that presumably should help to distinguish between species.

The database was created based on the eBird/Clements Checklist [30], but expanding to other checklists, especially those more regionally focused or based in other languages could provide additional insights. For example, the species officially known as the Jamaican Spindalis (*Spindalis nigricephala*) in the eBird/Clements 2022 Checklist has several local Jamaican names, including Mark Head, Cashew Bird, Silver Head, Spanish Quail, and Champa Beeza [69]. Future work could investigate how approaches to English-language common names vary regionally by comparing the

categorizations of species' officially recognized English-language names in the AvianLexiconAtlas to local, or alternative, English-language names on a regional scale. Lastly, the database is currently limited to the categorizations of common names of birds in American (U.S.) English, which is the standard language of the eBird/Clements checklist [34]. Historically, European and, later, American naturalists played outsized roles in the development of modern taxonomy that occurred alongside the expansion of Western imperialism around the world [1,16]. The relative distributions of categories for the unique descriptions of species English-language common names included in the database therefore mostly capture the philosophy of Western science in species nomenclature. Expanding the database and repeating the same methodology for common names in different languages and comparing the frequency of different types of descriptors used for the same species names across multiple languages would provide valuable insight into how approaches to avian nomenclature and the philosophy surrounding it vary across cultures. For example, some species might have many names in other languages in culturally diverse regions [70].

The AvianLexiconAtlas demonstrates that while common names of birds serve many purposes in English, there is a strong emphasis on descriptive characteristics associated with avian biology. The database highlights gaps in naming conventions, particularly for descriptors that are not associated with all individuals in a population or include specialized English-language terminology. We anticipate that further work using the database investigating common group names might find very different patterns, as there are different goals for the components of a name (i.e., to tell species apart or to identify groups of species). Furthermore, future work using the database could more specifically analyze different geographic and phylogenetic trends in common names in relation to the date of the species descriptions, where the species was first described (e.g., from a museum specimen or in the wild), the person the species description is attributed to, and types of life history or physical similarities across species. Investigations such as these would be useful for disentangling the influence of human history and avian biology on these observed patterns in avian nomenclature. The basic analyses describing the database we provided barely scratch the surface of what is possible for investigating trends in common names. Even beyond the myriad ways in which the database could be expanded, such as with additional taxonomies, languages, or groups of organisms, the AvianLexiconAtlas is a tool for researchers or amateur ornithologists alike. Researchers might associate the categorizations with other types of life history data (e.g., migratory patterns or sexual dichromatism), or investigate fine-scale analyses or specific clades or geographic locations. The amateur ornithologist might be curious about the history of a particular species name, and the AvianLexiconAtlas could help clarify its meaning or origins. This database is a rich resource that will enable a large variety of future work that can thus address the extent to which common names represent how people interact with birds today compared with historical interactions and across different cultural, biological, and regional contexts.

## Supporting information

**S1 Fig. Summary of mismatched category assignments in the 906 species common names assigned to different categories in Dataset A and Dataset B.**
(TIF)

**S2 Fig. Word cloud frequencies of terminology in common names.** Word cloud frequencies of (A) avian physical traits, (B) avian natural history traits, and (C) human-centered terminology. Names are scaled according to frequencies in each dataset.
(TIF)

**S3 Fig. Cladogram of the categories assigned to the English common names of 10,775 species of birds.** Categories of the English common names identified at the tips of the branches, based on color. Cladogram adapted from [38]. The inner circle includes species names associated with the general category of avian physical traits (both sexes physical trait, male physical trait, female physical trait, size). The middle circle includes species names associated with the general

category of avian natural history (behavior, geographic range, natural history), and the outer circle includes names associated with the general category of human-centered terminology unrelated to the biology of the species. See Table 1 for detailed explanations of each category.
(TIF)

**S1 Table. Calculation of D statistic for the phylogenetic structure of categories.** Results of Fritz & Purvis' [40] $D$ statistic calculations for the phylogenetic structure of each of the grouped categories: physical traits, natural history, and human-centered terminology. For each grouped category, species assigned to the category were represented by a state of 1 and the remaining species assigned to other categories were represented by a state of 0. $D$ is calculated by scaling the observed sum of sister-clade differences, $\Sigma d_{obs}$, with the mean values of the sum of sister-clade differences for 1,000 simulated trait distributions on the tips of the same phylogeny based on randomly reshuffling the trait values, $\Sigma d_r$, and trait evolution under Brownian motion $\Sigma d_b$: $D = [\Sigma d_{obs} - mean(\Sigma d_b)/[mean(\Sigma d_r) - mean(\Sigma d_b)]$. An estimated $D$ close to 1 represents a random distribution of a binary trait among related species on the phylogeny, while an estimated $D$ close to 0 represents a clumped distribution of a binary trait among related species that would be expected under the Brownian motion model of evolution. Calculations were completed using the R package *caper* 1.0.3 [41].
(PDF)

**S1 Appendix. AvianLexiconAtlas Database Files.** The data, glossary, and gazetteer reported in this article can be accessed at https://github.com/ajshultz/AvianLexiconAtlas.
(PDF)

## Acknowledgments

We thank *Birds of the World* for providing complimentary 1-week subscriptions for the student and amateur participants. We also thank Eden Bayou, Jacob Bijou, Jakiya J. Campbell, Emily Chen, Cindy Cheng, Reema Demopoulos, Savannah Garza, Tamar Hadad, Yujing He, Mia Hejlsberg, Jikke Inia, Yoonseo Jang, Zilai Jin, Risa Kanai, Nick Kruczynski, Langrun Li, Yujia Liu, Queenie Liu, Ziqi Ma, Nyjur Majok, Cecilia Méndez, Kimi Modiri, Yazmin Munoz, Iyioluwa Okediji, Sophia Ordonez, Mohammed Osmanu, Yian Pan, Wend-manegde Pitroipa, Stephanie Salas, Rylie Shaeffer, Andrew Shafer, Paul Shen, Jesse Sivan, Ciaran Timlin, Bryant To, Alexander Valenzuela-Jones, Frances Vandervoort, Nicholas Walsh, Qingyue Wang, Ruoxi Ye, Yawei Zhang, Guicheng Zhang, Yuchong Zhu, and Brian Zou for their contributions to the data collection. Thanks to Emily Webb, Kayce Bell, Jann Vendetti, the NHMLAC Urban Nature Research Center members, and two anonymous reviewers for providing comments on drafts of the manuscript.

## Author contributions

**Conceptualization:** Erin S. Morrison, Allison J. Shultz.

**Data curation:** Erin S. Morrison, Allison J. Shultz.

**Formal analysis:** Erin S. Morrison, Allison J. Shultz.

**Funding acquisition:** Erin S. Morrison.

**Investigation:** Erin S. Morrison, Guinevere P. Pandolfi, Stepfanie M. Aguillon, Jarome R. Ali, Olivia Archard, Daniel T. Baldassarre, Illeana Baquero, Kevin F.P. Bennett, Kevin M. Bonney, Riley Bryant, Rosanne M. Catanach, Therese A. Catanach, Ida Chavoshan, Sarah N. Davis, Brooke D. Goodman, Eric R. Gulson-Castillo, Matthew Hack, Jocelyn Hudon, Gavin M. Leighton, Kira M. Long, Dakota E. McCoy, J. F. McLaughlin, Ziqi Ma, Gaia Rueda Moreno, Talia M. Mota, Lara Noguchi, Ugo Nwigwe, Teresa Pegan, Kaiya L. Provost, Shauna A. Rasband, Jessie Frances Salter, Lauren C. Silvernail, Jared A. Simard, Heather R. Skeen, Juliana Soto-Patiño, Young Ha Suh, Qingyue Wang, Matthew E. Warshauer, Sissy Yan, Betsy Zalinski, Ziqi Zhao, Allison J. Shultz.

**Methodology:** Erin S. Morrison, Guinevere P. Pandolfi, Allison J. Shultz.

**Project administration:** Erin S. Morrison, Allison J. Shultz.

**Resources:** Erin S. Morrison, Allison J. Shultz.

**Supervision:** Erin S. Morrison, Allison J. Shultz.

**Validation:** Erin S. Morrison, Allison J. Shultz.

**Visualization:** Erin S. Morrison, Allison J. Shultz.

**Writing – original draft:** Erin S. Morrison, Allison J. Shultz.

**Writing – review & editing:** Erin S. Morrison, Stepfanie M. Aguillon, Daniel T. Baldassarre, Kevin F.P. Bennett, Kevin M. Bonney, Therese A. Catanach, Jocelyn Hudon, Gavin M. Leighton, J. F. McLaughlin, Kaiya L. Provost, Lauren C. Silvernail, Heather R. Skeen, Juliana Soto-Patiño, Allison J. Shultz.

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
