## [Decision Letter · Decision Letter 0]

31 Jan 2025

Dear Dr. Morrison,

Both reviewers commented that while they see the value of the AvianLexiconAtlas database, it is unclear whether the manuscript is meant to be a research article using the database, or a data paper which describes the database but does not test any specific research question with it. As this was submitted as a Research Article, following the traditional format of Introduction, Methods, Results and Discussion, the readers would expect to read a research paper that tests specific questions. Could the authors please clarify what their intentions are (i.e. is this meant to be a research article or a data paper), and if latter, refer to Guidelines for Specific Study Types on PLOSONE website and submit it under database category? Either way, I expect there to be a fair amount of restructuring and rewriting of the manuscript, hence the decision of major revision. Both reviewers provided a number of helpful and insightful suggestions whichever route the authors are going to take, so please refer to and consider making use of them.

We look forward to receiving your revised manuscript.

Kind regards,

Shoko Sugasawa

Academic Editor

PLOS ONE

Journal Requirements:

2. You indicated that ethical approval was not necessary for your study. We understand that the framework for ethical oversight requirements for studies of this type may differ depending on the setting and we would appreciate some further clarification regarding your research. Could you please provide further details on why your study is exempt from the need for approval and confirmation from your institutional review board or research ethics committee (e.g., in the form of a letter or email correspondence) that ethics review was not necessary for this study? Please include a copy of the correspondence as an ""Other"" file.

“Funding was provided by the New York University Liberal Studies New Faculty Scholarship Award to E.S.M”

Reviewers' comments:

Reviewer's Responses to Questions

**Comments to the Author**

1. Is the manuscript technically sound, and do the data support the conclusions?

Reviewer #1: Partly

Reviewer #2: Partly

2. Has the statistical analysis been performed appropriately and rigorously?

Reviewer #1: No

Reviewer #2: Yes

3. Have the authors made all data underlying the findings in their manuscript fully available?

Reviewer #1: No

Reviewer #2: Yes

4. Is the manuscript presented in an intelligible fashion and written in standard English?

Reviewer #1: Yes

Reviewer #2: Yes

Reviewer #1: This manuscript presents the ‘AvianLexiconAtlas’, a dataset of 10,906 species with their (American) English names classified into ten categories related to, e.g., the avian or cultural context of the bird. Basic descriptions of this data are presented, along with a long discussion about the coding decisions made when generating these crowd-sourced dataset. This is a really interesting dataset, and the manuscript itself is beautifully written. I only have a few technical comments (see below), which should be relatively easy to address, but I do have a broader, perhaps more philosophical point.

To my mind, this manuscript tries to be attempting several things at once, and thus both its aims and its impact is murky. If this is intended as a data paper, it does a great job, no further engagement with this paragraph required. (And, indeed, it’s possible that a subset of this author team has another manuscript in the works, but wants to get the dataset published first. Reasonable!) However, several other possible goals are perhaps attempted here.

• Is this manuscript testing hypothesis about this data? If so, what research question(s) is/are being addressed? This isn’t entirely clear to me. (More on this in the technical comments.)

• Is this manuscript presenting a case study on engaging undergraduates in the research process? If so, one would expect to see more engagement with the pedagogical literature. (Don’t read anything into this recommendation, other than that I recently saw it on BlueSky, but Stephan Lautenschlager recently published a piece of this ilk, https://doi.org/10.1186/s12052-024-00214-z )

• Is this manuscript trying to engage with the Bird Names for Birds movement? If so, I would expect to see slightly more connection to the motivation for changing bird names, and how past naming practices are being understood in modern times. (Again, don’t read too much into this recommendation, but Bond & Lavers recently had an Ibis paper doing just this, https://doi.org/10.1111/ibi.13356 )

• Is this manuscript trying to make a point about how humans relate to birds through naming conventions? If so, I would expect to see *far* more engagement with the linguistic / anthropological / ethno-ornithological literature on this topic. The work of the authors behind the EWA (Ethno-Ornithology World Atlas, https://ewatlas.net/ ) spring to mind, as does the vast literature on folk taxonomies. The discussion touches on this to a certain extent (e.g., L419-423, which are great) – I’d love to see ever more of this when setting up the context for this work. Similarly, I was fascinated by the terms in the gazette that needed to be defined, such as “hoary” or “poll” – the disconnect in vocabulary between ornithologists and the general public is an excellent point!

Anyway, all of this is mostly a scope / editorial point rather than a reviewer point, but I figured it might be useful context for the authors in how an audience might read and interpret their work. Again, I want to stress that I really enjoyed reading this – it’s a cool dataset! – and if the authors have anything else planned for this topic, I will greatly look forward to reading it.

Technical comments:

L91: It’s not clear to me what “As part of these deliberations” refers to.

L101-104: Why would you predict this? Much more context is needed to understand why there might be geographic and/or taxonomic patterns towards naming conventions.

L104-106: What insights could this provide? This context is currently missing for the reader.

L147 onward: This is possibly an unpopular opinion, but I found this description of the 11-versus-10 category thing very hard to follow. It would perhaps be less confusing to either straight-up omit the mention of the 11 categories, or to say something like “Though 11 categories were initially scored, these were further refined during the data cleaning phase to a set of 10 categories [describe the 10]”.

L152: This may be rendered irrelevant by the above comment, but the use of “random” here is distracting, because of course birds do not have “random” names. They may have names that are miscellaneous, or opaque, or ‘other’, or something, but language is not ‘random’.

Paragraph beginning L165: Were all data collectors fluent speakers of English? (Obviously for an undergraduate class, participants could not be excluded based on language ability, but it’s useful context for a language-based coding task.)

Similarly, for the scorers who made assessments based on photographs, did all scorers have typical colour vision (e.g., the ability to distinguish between red and green)?

L180: Comma needed after the bracket/parenthesis.

L230: Again, why would you predict this? This needs to be spelled out somewhere (e.g., the introduction, or maybe the supplement).

L234: My understanding is that this phylogeny (and the associated R package) is currently unreviewed. I don’t think *I* care (there’s a trade-off to be had between including nearly all species in an unreviewed phylogeny and, say, losing ~10% of the species to use a published phylogeny with known biases, and I’m sympathetic to either decision), but the editor may have an opinion.

Paragraph beginning L247: Why was this done as a chi-squared test and not with a phylogenetic correction? You know that there’s a phylogenetic signal in the data, and that species naming conventions are non-independent – you should be correcting for that.

L248: How did you extract the general region? How is “general region” defined? Also, is this in the published data and I just can’t find it in the Github?

Table 2: It took me a very long time to understand “Mean times repeated if not unique” and “Median times repeated if not unique”. Please consider clarifying somewhere.

Figure 2: How was this word cloud made? (What software/package?) (Apologies if I missed this!)

L330: These differences in D values are relatively small; without confidence intervals, the reader doesn’t know how to distinguish them. I’d suggest either adding confidence intervals, or softening the phrasing to “somewhat stronger” (or something).

A lot of your discussion is really great. For example, your point about the naming of Grallaria species is a lovely insight, as is the idea that closely-related seabird species are more distinguishable by geography than by morphology. I’d, personally, love to see even more about the (highly colonial) history of how people have named birds in (American) English and how that interfaces with global patterns of, e.g., sexual dimorphism, community assembly, even if it’s just anecdotal.

L485-486: This strikes me as an odd question, given that this manuscript presents data that can, in part, answer this question. Perhaps rephrase? In particular, what scientific * question * (ornithological, linguistic, ethno-ornithological, whatever) would be addressed by conducting an investigation like this?

(Again, it almost feels like the authors are queuing up a follow-up study, with more anthropological/linguistic insights. If so, of course, that’s fine, but maybe dialing a little bit back on that and focusing a little more about the impact of * this * study would help improve the stand-alone effect of this work on the field.)

Table S2: The last three rows of Table S2 are very difficult to understand unless you dig into Fritz & Purvis 2010 and look up the definitions of things like \sum d_{obs} . Consider either omitting or explaining somewhere how D is actually calculated?

Gazette:

• There are several changes in font and font size that you might want to fix.

• You probably want to remove the note in red on the first page.

• Do you have permission to publish these figures?

• You might want to triple-check all of the political affiliations claimed in the geographic location section, and/or include a disclaimer; these in general seem excellent, but people have strong feelings about all sorts of things.

• The Perijá entry is quite obviously copied from Wikipedia – you may want to disable the links and standardize the font, if nothing else.

Reviewer #2: In this study, authors describe an AvianLexiconAtlas which contains assignments for descriptive English-language common names of 10,906 birds. Authors state that there is currently no systematic assessment of how often common names communicate identifiable and biologically relevant characteristics about species, suggesting that this is an issue in ornithology because common names are used more often than scientific names even by professional researchers. Hence, through a collaboration of professional ornithologists, amateurs, and students they classify these names into 10 categories based on avian physical traits, natural history and human-constructed terminology. They show that 89% of the birds are named after some aspect of their biology, either after a physical trait found in one or both sexes or after their natural history or behaviour. When birds are named after some aspect of their biology, name descriptors appear to be geographically and phylogenetically clustered.

The manuscript is clearly written and relatively easy to read, however, I have some concerns about what the main aim of the study is and the take home message it intends to convey.

In addition to making a clear list of achievable objectives in the introduction which should be reflected in the methods, results and discussions, I have the following feedback for consideration by the authors:

1. I struggled to identify the main biological problem addressed in the manuscript or the main biological insight gained. I can understand the fact that name descriptors based on biology are more informative and that some name descriptors are very localised and so not very helpful for generalisation. I also understand that there are debates about fair naming – what bird common names shouldn’t (or should) be. Is this analysis in light of these debates? Otherwise, if the main aim is to show how bird common names are arrived at or the unique histories of such, then the manuscript could be better focused, with that aim clearly stated, and more involving. There are some predictions outlined in the introduction and methods, like those associated with phylogenetic and geographical clustering of names (lines 100 – 104 and 230 - 231), but there isn’t a clear indication of what might be the case if these predictions were right. Moreover, it might be important to show whether there are any biological implications for common names that do not follow accepted naming conventions (if any).

2. Authors have taken time (e.g line 273 - 275) to explain author contributions and criteria for qualifying for co-authorship. This is useful in some ways, but I do not believe that this is relevant for the main text. It might be required by the journal to judge fair authorship, but I believe that it should be included in the author contribution section, and not in the main text. If the inclusion or exclusion criteria has an impact on the results or the interpretation of it, then maybe these criteria discussed.

3. Line 109: it may be a goal of common names that they are useful for the general public, but they must not be generalised to the extent that they cease to make sense locally, except a species species means exactly the same thing to everyone, and I do not believe that this is the case. An interesting perspective for me would be to discuss how names differ according to context or locations (see you comment in lines 512 - 513).

Specific comments

Lines 405 – species not specious.

The term human-constructed terminology is a bit misleading because all names are human-constructed including avian centred names.

Lines 349 – 351 – your result which shows that Africa and Oceans show the highest proportions of species named after human-constructed terminology seemed to have been ignored in the discussion. Isn’t it easily predictable that species described by colonial ornithologists (which may be the case for most of Africa and the Oceans) are more likely to be named after persons? You mentioned the outsized roles of western naturalists in lines 502 – 508. Although, I am curious about why South America has the least proportion of names based on human-constructed terminologies. The timing of species description may influence how they are described. Newly described species are less likely to have been described from a museum collection which you argue may have been the motivation for naming species based on physical traits.

Lines 417 – 501 discusses methodological considerations and participation. Down-sizing this aspect of the discussion may help make the manuscript better concise.

Good luck with the revisions.

**Do you want your identity to be public for this peer review?** For information about this choice, including consent withdrawal, please see our Privacy Policy

Reviewer #1: No

Reviewer #2: No

---

## [Author Response · Author response to Decision Letter 1]

19 Mar 2025

Dear Dr. Sugasawa,

Thank you and the two anonymous reviewers for the helpful feedback on our manuscript; we believe the comments have resulted in a considerably strengthened manuscript. We appreciate the opportunity to submit a revised version of the manuscript based on the reviews we received. Based on the main feedback we received from you and both reviewers, we have now reframed the manuscript to be a Research Article that solely reports on a new database, the AvianLexiconAtlas. We are no longer using the manuscript to present the results of original research that address a clearly defined research question. As part of this database description, we now include a section in the Results (lines 299-315) focusing on the implementation of the database that specifically provides a direct link to the database hosting site, plans for long-term database maintenance and growth. We retained the analyses included in the initial submission, but now provide support that these analyses are simply a way to validate the utility of the database. The trends that are presented in these analyses can be used to ask future questions about the utility of these English-language descriptors in a biological context as well as provide data for further examination into the linguistic history of these descriptors. In the Discussion we use these trends as a way to propose several areas of future research using the database.

As part of the revision, we have provided further details on why this study was exempt from the need for approval from the institutional review board. In the ‘Data Collectors’ section of the Methods (lines 194-201), we now include the full ethics statement. We specifically explain that the New York University (NYU) Human Research Protection Program (HRPP) determined via verbal and written consent that the work involved in the data collection for the database did not meet the criteria involving human subjects per the United States’ regulations for the protection of human subjects (45CFR46.102(e)). This determination was made because the focus of the research was not on collecting information about the data collectors, but only involved a crowdsourcing model where the data collectors helped categorize the meanings behind unique descriptors in bird names. As such, the data collection methods did not require HRPP review and approval. We have included an official letter from the NYU Human Research Protection Program also stating this language in the revision documents. This has been submitted as an “Other” file. To further clarify that this study is a crowdsourced data collection research project associated with scientific nomenclature, and not a study on human subjects, we have changed from calling the individuals who helped with categorizing the bird names ‘participants’ and now call them ‘data collectors’ throughout the entire manuscript.

As per the journal’s request, we now have an amended Role of Funder statement, and thank the journal editorial staff for changing this on our behalf:

“Funding was provided by the New York University Liberal Studies New Faculty Scholarship Award to E.S.M. The funders had no role in study design, data collection and analysis, decision to publish, or preparation of the manuscript.”

Please note, that due to the removal of the S1 Table and the S1 Figure from the revised manuscript, the numbering of the remaining Supporting Information table and figures has now been changed. This has been updated in the manuscript, and we describe why this table and figure were removed from the Supporting Information documentation in detail below. We also edited the text for further clarity and PLOS ONE style requirements

Below is the list of specific changes incorporated in the revised manuscript:

1. Reviewer 1: L91: It’s not clear to me what “As part of these deliberations” refers to

Authors: This phrase has been deleted and in its place (lines 115-117) we have now clarified that the goal of constructing this database is to be able to understand what common names currently communicate about species in terms of the scope and variability of the terminology.

2. Reviewer 1: L101-104: Why would you predict this? Much more context is needed to understand why there might be geographic and/or taxonomic patterns towards naming conventions.

Reviewer 1: L104-106: What insights could this provide? This context is currently missing for the reader.

Reviewer 2: I struggled to identify the main biological problem addressed in the manuscript or the main biological insight gained. I can understand the fact that name descriptors based on biology are more informative and that some name descriptors are very localised and so not very helpful for generalisation. I also understand that there are debates about fair naming – what bird common names shouldn’t (or should) be. Is this analysis in light of these debates? Otherwise, if the main aim is to show how bird common names are arrived at or the unique histories of such, then the manuscript could be better focused, with that aim clearly stated, and more involving. There are some predictions outlined in the introduction and methods, like those associated with phylogenetic and geographical clustering of names (lines 100 – 104 and 230 - 231), but there isn’t a clear indication of what might be the case if these predictions were right. Moreover, it might be important to show whether there are any biological implications for common names that do not follow accepted naming conventions (if any).

Authors: The predictions in the original manuscript in L101-104 have been deleted from the introduction to align with the focus of the paper on introducing the database. Since we are no longer presenting the work as a hypothesis test, we did not believe we should retain the predictions, which we believe led to some of the confusion about what the goal of this manuscript was. Instead, the introduction of phylogenetic and biogeographic analyses in the Introduction section (lines 125-132) are presented as a way to validate the utility of the dataset for future research questions, and we propose several future research topics in the Discussion as part of our initial observations about the trends we observed in the database (lines 442-453).

3. Reviewer 1: L485-486: This strikes me as an odd question, given that this manuscript presents data that can, in part, answer this question. Perhaps rephrase? In particular, what scientific * question * (ornithological, linguistic, ethno-ornithological, whatever) would be addressed by conducting an investigation like this?

Authors: We have added a line after this question to clarify its significance (lines 499-502).

4. Reviewer 1: L147 onward: This is possibly an unpopular opinion, but I found this description of the 11-versus-10 category thing very hard to follow. It would perhaps be less confusing to either straight-up omit the mention of the 11 categories, or to say something like “Though 11 categories were initially scored, these were further refined during the data cleaning phase to a set of 10 categories [describe the 10]”.

Authors: Thank you for pointing this out, we also can see how this could be confusing and per your advice have simplified the initial description of the categories in the Methods section (lines 171-177) and omit the mention of the 11 initial categories. We now only present the set of 10 categories used for terminology assignments. As part of this update, we moved the more detailed descriptions of each of the 10 categories into Table 1 that were previously only included in S1 Table. Since the S1 Table only existed to provide a comparison between the 11 initial categories and the 10 final categories we decided to delete the S1 Table about these 11 categories from the manuscript.

5. Reviewer 1: L152: This may be rendered irrelevant by the above comment, but the use of “random” here is distracting, because of course birds do not have “random” names. They may have names that are miscellaneous, or opaque, or ‘other’, or something, but language is not ‘random’.

Authors: We have addressed this issue by removing any mention of the initial 11 categories from the manuscript.

6. Reviewer 1: Paragraph beginning L165: Were all data collectors fluent speakers of English? (Obviously for an undergraduate class, participants could not be excluded based on language ability, but it’s useful context for a language-based coding task.)

Reviewer 1: Similarly, for the scorers who made assessments based on photographs, did all scorers have typical colour vision (e.g., the ability to distinguish between red and green)?

Authors: We have now added a clarifying sentence in the ‘Data collection’ subsection (lines 207-210) of the Methods that explains the use of duplicate datasets (in the first round of scoring) and triplicate datasets (in the second round of scoring) in the design of the data collection was established to validate the accuracy of the category assignments made by different people. We decided not to specifically address differences in English language fluency and/or color vision of the data collectors in the manuscript, because we did not ask any of the data collectors about this during the recruitment process.

7. Reviewer 1: L180: Comma needed after the bracket/parenthesis.

Authors: The comma has now been added in line 204.

8. Reviewer 1: L230: Again, why would you predict this? This needs to be spelled out somewhere (e.g., the introduction, or maybe the supplement).

Authors: Based on the decision to reframe the manuscript as a description of a new database, we have now changed the language in this section of the Methods and removed the mention of ‘predictions’ for the analyses used in the manuscript. Instead, this section is now called ‘Data validation’ (line 258) and we now described the analyses as a way to validate the utility of the database by simply exploring trends in the dataset that was collected (lines 259-296). This goal for the analyses has also been updated in the Introduction and Discussion sections to remove the implication we are hypothesis testing with these analyses.

9. Reviewer 1: My understanding is that this phylogeny (and the associated R package) is currently unreviewed. I don’t think *I* care (there’s a trade-off to be had between including nearly all species in an unreviewed phylogeny and, say, losing ~10% of the species to use a published phylogeny with known biases, and I’m sympathetic to either decision), but the editor may have an opinion.

Authors: The decision to use the unpublished phylogeny that is currently in review was made because it most closely aligns with the taxonomy of the species in the database, because they are both based on the eBird/Clements taxonomy. This phylogeny, its documented methodology, and its associated R programs are all currently available as a preprint on BioRxiv (McTavish et al., https://www.biorxiv.org/content/10.1101/2024.05.20.595017v1). Additionally, as of the date of this letter the phylogeny has been cited by 4 published peer reviewed studies:

• Barber, R.A. et al. (2024). PLOS Biol. 22(11): e3002856. https://doi.org/10.1371/journal.pbio.3002856

• Janzen, E. & Etienne, R.S. (2024). Mol Phylogenet Evol. 200: 108168. https://doi.org/10.1016/j.ympev.2024.108168

• Nussbaumer, R. et al. (2025). Divers Distrib. 31: e13935. https://doi.org/10.1111/ddi.13935

• Van Doren, B.M. et al. (2025). Curr Biol. 35(4): P898-904.E4. https://doi.org/10.1016/j.cub.2024.12.033

10. Reviewer 1: Paragraph beginning L247: Why was this done as a chi-squared test and not with a phylogenetic correction? You know that there’s a phylogenetic signal in the data, and that species naming conventions are non-independent – you should be correcting for that

Authors: Since several species did not have unique values for breeding locations, a phylogenetic regression analysis or a linear model was not possible. We agree that there may be some tendencies for species from the same families and orders to be named in a similar manner from the same geographic region, but in this manuscript we do not look to completely explain these patterns and we believe that question is now beyond the scope of the revised manuscript. Thus, we felt that the chi-squared test is describing patterns that may intrigue readers enough to investigate it more thoroughly elsewhere.

11. Reviewer 1: L248: How did you extract the general region? How is “general region” defined? Also, is this in the published data and I just can’t find it in the Github?

Authors: We provided more details in the “Data validation” section of the Methods to clarify what we mean by “extracted” the general region (lines 284-286). This data is compiled by the IOC World Bird List and we extracted the specific general regions for each species in the IOC World Bird list. We retained the definition for “general region” that was included in the initial submission of the manuscript, which describes the term as geographic ranges at the continent level, but we have now added a reference to the IOC World Bird List Range Terminology website so the reader can learn more details about how these designations were made. We have added the the general region or regions used for the breeding range of each species from the IOC World Bird List 13.2 to the ‘total_dataset_allrounds.csv’ file posted on the Github site for the database (S1 Appendix, https://github.com/ajshultz/AvianLexiconAtlas).

12. Reviewer 2: Table 2: It took me a very long time to understand “Mean times repeated if not unique” and “Median times repeated if not unique”. Please consider clarifying somewhere.

Authors: We have added text in the legend of Table 2 that clarifies that calculations for the mean and median frequencies of unique descriptors only include descriptors that occur more than once within species names in each category. Additionally, we changed the titles of the columns associated with the mean and median times to: “Mean frequency of unique descriptors” and “Median frequency of unique descriptors”

13. Reviewer 1: Figure 2: How was this word cloud made? (What software/package?)

Authors: We have now identified the R software package (wordcloud2) used to produce the word clouds in the “Data validation” section of the Methods (lines 262-263).

14. Reviewer 1: L330: These differences in D values are relatively small; without confidence intervals, the reader doesn’t know how to distinguish them. I’d suggest either adding confidence intervals, or softening the phrasing to “somewhat stronger” (or something).

Authors: The wording to describe the differences in the D values has been changed to “somewhat stronger” (line 358). There are no confidence intervals as part of the calculation for D, and we agree with Reviewer 1 that it is difficult to distinguish the magnitude of difference in this case

15. Reviewer 1: Table S2: The last three rows of Table S2 are very difficult to understand unless you dig into Fritz & Purvis 2010 and look up the definitions of things like \sum d_{obs}. Consider either omitting or explaining somewhere how D is actually calculated?

Authors: Due to the deletion of the table that was originally S1 Table from the revised manuscript, the supporting information table reporting the calculations for the D statistic that was S2 Table in the original manuscript has now been changed to S1 Table in the revised manuscript. The legend of this table has now been expanded to provide more detail on how D is actually calculated. It includes the specific equation used for D along with clarifying definitions for each of the calculations and statistics reported in the table.

16. Reviewer 1:

Gazette:

• There are several changes in font and font size that you might want to fix.

• You probably want to remove the note in red on the first page.

• Do you have permission to publish these figures?

• You might want to triple-check all of the political affiliations claimed in the geographic location section, and/or include a disclaimer; these in general seem excellent, but people have strong feelings about all sorts of things.

• The Perijá entry is quite obviously copied from Wikipedia – you may want to disable the links an

---

## [Decision Letter · Decision Letter 1]

30 Apr 2025

Dear Dr. Morrison,

We look forward to receiving your revised manuscript.

Kind regards,

Shoko Sugasawa

Academic Editor

PLOS ONE

Journal Requirements:

Reviewers' comments:

Reviewer's Responses to Questions

**Comments to the Author**

Reviewer #1: (No Response)

Reviewer #2: (No Response)

2. Is the manuscript technically sound, and do the data support the conclusions?

Reviewer #1: Yes

Reviewer #2: Partly

3. Has the statistical analysis been performed appropriately and rigorously?

Reviewer #1: Yes

Reviewer #2: N/A

4. Have the authors made all data underlying the findings in their manuscript fully available?

Reviewer #1: Yes

Reviewer #2: Yes

5. Is the manuscript presented in an intelligible fashion and written in standard English?

Reviewer #1: Yes

Reviewer #2: No

Reviewer #1: I thank the authors for their thorough and thoughtful revisions, which have greatly increased the readability of this manuscript. The current version of this work is very interesting, and I enjoyed the opportunity to read it.

I have just a few extremely minor comments remaining:

L74: They’re not necessarily Latin. “Latinized”? “binomial, often in Latin”? Something else?

L103: Perhaps the “gull” family of birds? Or just “gull species”?

L238: Purvis, not Pervis

L248-249: What are some other general regions, if not continents?

L537-538: This is extremely picky, but I’d recommend phrasing this sentence with a little less surprise; the fact that colour naming varies among people is an extremely well-documented anthropological phenomenon (e.g., https://doi.org/10.1073/pnas.2109237118 , or even Berlin & Kay’s “Basic Color Terms”)

Figures – as a heads-up, Manuscript Central (or whatever software this is – “Editorial Manager”??) has compressed your figures in a weird way. It’s also obliterated the mathematical formatting on Page 35.

Gazetteer:

• Extra square bracket under ‘Dimorphic’

• Extra bolding in ‘pinnated’, ‘russet’, ‘tawny’

• I don’t know what “If toponym varies from spelling indexed under, listed at end of entry” means?

Reviewer #2: Thank you for revising your manuscript and responding the comments made on the initial draft. I have made some recommendations to the Editor who will advise you appropriately on how the manuscript might revised further as a data paper.

**Do you want your identity to be public for this peer review?** For information about this choice, including consent withdrawal, please see our Privacy Policy

Reviewer #1: No

Reviewer #2: No

---

## [Author Response · Author response to Decision Letter 2]

15 May 2025

Dear Dr. Sugasawa,

Thank you and the two anonymous reviewers for taking the time to review our revised manuscript and providing further feedback. We appreciate the opportunity to continue to revise the manuscript, and believe we have thoroughly addressed the comments from the reviewers.

As part of the first revision, we provided further details on why this study was exempt from the need for approval from the institutional review board. As requested by the journal, we have included an Ethics statement subsection in lines 145-153 at the start of the Materials and methods section. We specifically explain that the New York University (NYU) Human Research Protection Program (HRPP) determined via verbal and written consent that the work involved in the data collection for the database did not meet the criteria involving human subjects per the United States’ regulations for the protection of human subjects (45CFR46.102(e)). This determination was made because the focus of the research was not on collecting information about the data collectors, but only involved a crowdsourcing model where the data collectors helped categorize the meanings behind unique descriptors in bird names. As such, the data collection methods did not require HRPP review and approval. We have included an official letter from the NYU Human Research Protection Program also stating this language in the revision documents. This has been submitted as an “Other” file. To further clarify that this study is a crowdsourced data collection research project associated with scientific nomenclature, and not a study on human subjects, we have changed from calling the individuals who helped with categorizing the bird names ‘participants’ and now call them ‘data collectors’ throughout the entire manuscript.

As per the journal’s request, we now have an amended Role of Funder statement, and thank the journal editorial staff for changing this on our behalf:

“Funding was provided by the New York University Liberal Studies New Faculty Scholarship

Award to E.S.M. The funders had no role in study design, data collection and analysis, decision

to publish, or preparation of the manuscript.”

Below is the list of specific changes incorporated in the revised manuscript:

1. Reviewer 2 & Academic Editor: Reviewer2 commented that the discussion section should clarify the database utility and prospects, which I agree. As the current discussion is structured around the results and the constraints of the database, the significance and future ideas for the database are scattered across the discussion, making it hard to get a comprehensive and concrete idea of how this database contributes to research in biology and other fields. Please clarify the utility and future prospects of the database in discussion -- I think that rewriting the concluding paragraph might be the least destructive way, but am open to other ways to achieve this.

Authors: We appreciate the comments by Reviewer 2 and the Academic Editor, and have revised the Discussion as suggested. We now have restructured the end of the discussion to be centered on the utility and future prospects of the database. We have reframed the constraints as opportunities to expand the database in the future, summarize all of the opportunities for future research directions mentioned throughout the Discussion, and end on highlighting the significance of the AvianLexiconAtlas for both professional researchers and amateurs.

2. Reviewer 1:

L74: They’re not necessarily Latin. “Latinized”? “binomial, often in Latin”? Something else?

L103: Perhaps the “gull” family of birds? Or just “gull species”?

L238: Purvis, not Pervis

Authors: In line 100, we have edited this passage to now say that ICZN scientific names are written as a binomial, often in Latin. In line 276 we corrected the spelling error and it now says Purvis.

The topics of the lines referenced by Reviewer 1 don’t match the ones in the manuscript file we submitted in the first revision. In the case of the referenced line 103, we were not entirely sure what this was in reference to, as we don’t mention any gull species in the manuscript. We thought the comment might be referring to the use of “shared group names” to describe the part of a species’ English common name that is shared across species (for example Ostrich) in lines 160-165 in the manuscript for the first revision. In the current version of the manuscript, we have now deleted the term “group” in lines 172-177, and instead refer to the names that occur in multiple species, as “shared names”. We chose not to specifically define these shared names as ‘family’ names, because the shared descriptor across species may not always correspond directly to taxonomic groups.

3. Reviewer 1: L248-249: What are some other general regions, if not continents?

In lines 288-291 we have now clarified that the general range classifications for species occur at the subcontinent level (instead saying that the classifications were generally, but not always, at the continent level) and have added that in addition to these regions there is a separate classification for primarily oceanic species.

4. Reviewer 1: L537-538: This is extremely picky, but I’d recommend phrasing this sentence with a little less surprise; the fact that colour naming varies among people is an extremely well-documented anthropological phenomenon (e.g., https://doi.org/10.1073/pnas.2109237118 , or even Berlin & Kay’s “Basic Color Terms”)

Authors: Thank you for bringing this work to our attention. We have now incorporated both of these sources into the manuscript in lines 474-478 to hypothesize that the diversity of color terms observed in the dataset are likely associated with how often general categories of color are referenced across species. This section about color terminology was moved from the concluding paragraph of the Discussion to an earlier paragraph in the Discussion that provides an overview of the issues data collectors faced with unfamiliar terminology.

5. Reviewer 1: Figures – as a heads-up, Manuscript Central (or whatever software this is – “Editorial Manager”??) has compressed your figures in a weird way. It’s also obliterated the mathematical formatting on Page 35.

Authors: Thank you for bringing this to our attention. We have been in touch with the PLOS ONE editorial office and were told that “the reduction in resolution is part of the PDF building process that cannot be prevented, and the PDF version of manuscripts will always contain a compressed version of very high-resolution figures.” The editorial office advised us that all of the files are available to download in their original format in the manuscript’s file inventory. We tested this and the files and images were viewable in the form we intended.

6. Reviewer 1:

Gazetteer:

• Extra square bracket under ‘Dimorphic’

• Extra bolding in ‘pinnated’, ‘russet’, ‘tawny’

• I don’t know what “If toponym varies from spelling indexed under, listed at end of entry” means?

Authors: The extra square bracket and extra bolding in those entries have now been removed. We have now reworded the phrase “If toponym varies from spelling indexed under, listed at end of entry” that was located at the start of the Gazetteer to clarify its meaning, and it is now phrased as: “Index of specific place names. Any documented variations of the spelling for an indexed place name are listed at the end of its entry.” The new version of the Glossary and Gazetteer file has been uploaded to the GitHub site.

7. Authors: In the first round of feedback we received, there was concern that the phylogeny referenced in the manuscript was currently unreviewed. In the time since we submitted the first revision, however, the phylogeny has now been peer reviewed and published in PNAS:

McTavish EJ, Gerbracht JA, Holder MT, Iliff MJ, Lepage D, Rasmussen PC, et al. A complete and dynamic tree of birds. Proc Natl Acad Sci. 2025;122(18):e2409658122. https://doi.org/10.1073/pnas.2409658122

We have now updated the citation (38) for the phylogeny in the revised manuscript to its peer reviewed publication.

Thank you, again, to all of the reviewers for your continued feedback.

Sincerely,

Dr. Erin Morrison and Dr. Allison Shultz

---

## [Editor Report · Decision Letter 2]

21 May 2025

AvianLexiconAtlas: A database of descriptive categories of English-language bird names around the world

PONE-D-24-54740R2

Dear Dr. Morrison,

We’re pleased to inform you that your manuscript has been judged scientifically suitable for publication and will be formally accepted for publication once it meets all outstanding technical requirements.

Kind regards,

Shoko Sugasawa

Academic Editor

PLOS ONE

---

## [Editor Report · Acceptance letter]

PONE-D-24-54740R2

PLOS ONE

Dear Dr. Morrison,

I'm pleased to inform you that your manuscript has been deemed suitable for publication in PLOS ONE. Congratulations! Your manuscript is now being handed over to our production team.

Kind regards,

on behalf of

Dr. Shoko Sugasawa

Academic Editor

PLOS ONE